# RACH-Space: Reconstructing Adaptive Convex Hull Space with applications in weak supervision

## Abstract

We introduce *RACH-Space*, a novel classification method in ensemble learning. In particular, we show its applicability as a label model for weakly supervised learning. *RACH-Space* offers simplicity in implementation with minimal assumptions on the data or weak signals. The model is well suited for scenarios where fully labeled data is not available. Our method is built upon geometrical interpretation of the space spanned by weak signals. Our analysis of the high dimensional convex hull structure underlying general set of weak signals bridges geometry with machine learning. Empirical results also demonstrate that *RACH-Space* works well in practice and compares favorably to best existing label models for weakly supervised learning.

## 1 Introduction

In machine learning applications, the preparation of labeled data poses a major challenge. Given this, some researchers have explored *weakly supervised learning*. This approach involves the integration of inexpensive, noisy signals that provide partial information regarding the labels assigned to each data point. By combining these signals, a synthetic label is generated for the raw dataset. These signals are far from perfect, as they only provide partial information about the data points, and sometimes abstain and give incomplete information about the dataset as a whole. Hence, they provide *"weak"* supervision for labeling the data points. They come from diverse resources such as heuristics (Shin et al., 2015) and knowledge bases (Mintz et al., 2009).

The setup for weakly supervised learning is as follows. A dataset comprising of $n$ data points is given, and the task is to classify them into $k$ classes. Let $\mathbf{X}$ represent the set of points, and $\mathbf{y} \in \{0, 1\}^{nk}$ denote the ground-truth label vector, where entries of $1$ or $0$ indicate class membership or non-membership, respectively. Formally, the $(k-1)n+i$-th entry of $\mathbf{y}$ signifies whether the $i$-th data point belongs to class $k$. In weak supervision, $m$ weak signals $\mathbf{w}_1, \mathbf{w}_2, ..., \mathbf{w}_m$ are provided, where each entry of $\mathbf{w}_i \in [0, 1]^{nk}$ represent the probability that a data point belongs to a specific class. All weak signals are collected to form the matrix $\mathbf{W} \in [0, 1]^{m \times nk}$, where the $i$-th row corresponds to the $i$-th weak signal, and the $(i, (k-1)n+j)$-th entry indicates the probability of the $j$-th data point belonging to class $k$. The main problem in weakly supervised learning is stated as follows:

**Problem**: Consider a setup with $n$ training data points $m$ weak signals. These signals provide partial information about classifying each data point into $k$ classes. The task is to find a label that provides the "best" possible approximation of the unknown ground truth label, $\mathbf{y}$.

In this paper, motivated by the approach in Arachie & Huang (2021), we address this problem by considering the expected empirical error rate of the weak signal. We first introduce the expected empirical error rate of a weak signal $\mathbf{w_i}$, denoted as $\epsilon_i$. $\epsilon_i$ is given by:

$$\epsilon_i = \frac{1}{nk}(\mathbf{w_i}^\top(\mathbf{1} - \mathbf{y}) + (\mathbf{1} - \mathbf{w_i})^\top \mathbf{y}) = \frac{1}{nk}(\mathbf{1}^\top \mathbf{y} - 2\mathbf{w_i}^\top \mathbf{y} + \mathbf{w_i}^\top \mathbf{1}). \tag{1.1}$$

Given a single ground-truth classification for each of the $n$ data points, i.e., $\mathbf{1}^\top \mathbf{y} = n$, the expected empirical error rate simplifies to:

$$\epsilon_i = \frac{1}{nk}(-\mathbf{2w_i}^\top \mathbf{y} + \mathbf{w_i}^\top \mathbf{1} + n). \tag{1.2}$$

If the expected empirical rates are known, the above equation can be formulated as a linear system $\mathbf{Ay} = \mathbf{b}$ where $\mathbf{A} = 2\mathbf{W}$, where each row of $\mathbf{W}$ corresponds to weak signals and $\mathbf{b} \in \mathbb{R}^m$ with $b_i = -nk \cdot \epsilon_i + \mathbf{w_i}^\top \mathbf{1} + n$. However, as the expected empirical error rates are unknown, this formulation cannot be solved without requiring strong assumptions about prior knowledge of the weak signals. In this paper, we introduce a new framework called *RACH-Space*. Unlike Arachie & Huang (2021), RACH-Space does not assume prior knowledge of expected empirical error rates of the weak signals. The key idea is to modify $\mathbf{b}$ by introducing a pseudo-error rate as follows. Formally, we define $\widetilde{\mathbf{b}} \in \mathbb{R}^m$ with $\widetilde{b}_i = -nk \cdot \epsilon + \mathbf{w_i}^\top \mathbf{1} + n$, where $\epsilon$ is a pseudo-error rate defined as the expected empirical error rate of a weak signal that is better than random classification of the data points (see Lemma 3.1) i.e., a labeling whereby the probability that a data point is in any given class is $\frac{1}{k}$. Notably, $\widetilde{\mathbf{b}}$ is initially defined using $\epsilon$ and is subsequently refined by incrementally adjusting its value. A key aspect of *RACH-Space* involves adjusting the parameter $\widetilde{\mathbf{b}}$. The goal is to position $\frac{\widetilde{\mathbf{b}}}{n}$ within a *safe region*, defined within the convex hull that is formed by the weak signals. An illustration is shown below.

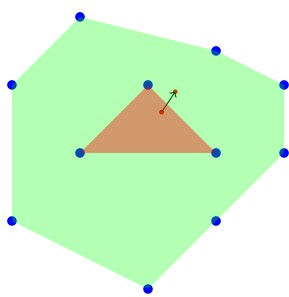

Figure 1: An illustration for the main idea of *RACH-Space*. Using an initial pseudo-error rate $\epsilon$, the right hand side vector $\widetilde{\mathbf{b}}$ is defined. $\frac{\widetilde{\mathbf{b}}}{n}$ is initially in the inner convex hull and our aim is to update the pseudo-error rate such that $\frac{\widetilde{\mathbf{b}}}{n}$ lies in the *safe region*, which is the region inside the convex hull but outside the second largest convex hull.

We want to emphasize that our primary algorithm updates the parameter $\widetilde{\mathbf{b}}$ based on the pseudo-error rate, without requiring explicit prior knowledge about the error rates of weak signals nor requiring strong assumptions about actual values of empirical error rates of weak signals.

**Contributions.** The contributions of this paper are as follows:

1. Proposing *RACH-Space*, an efficient label model that provides synthetic labels for the raw dataset when there are partial, incomplete supervision of the data points.

2. Designing *RACH-Space* to work on simple assumptions, not requiring any prior knowledge of the data or the weak signals nor any particular distribution of the weak signals. This allows it to be successful in different types of classification tasks.

3. Connecting high dimensional geometry with machine learning. *RACH-Space* first identifies the extreme points of Conv(Col($\mathbf{A}$)) correspond to parts of the weak signals $\mathbf{W}$ with the strongest class indication. Then we introduce the concept of *safe region*, the space between the convex hull and the second largest convex hull.

4. Providing theoretical evidence that $\frac{\widetilde{\mathbf{b}}}{n}$ should be in *safe region* to avoid the synthetic label $\widetilde{\mathbf{y}}$ ignoring the strongest class indication of weak signals. We also provide empirical evidence on real world datasets that supports the significance of this finding as well.

5. We show that on the 14 real world datasets provided by the *WRENCH* benchmark for weak supervision (Zhang et al., 2021) where it is common to face weakly supervised learning problems, *RACH-Space* outperforms the best existing label models, including state-of-art results for 4 of the dataset.

In section 2, we commence by discussing pertinent literature that provided the motivation for our algorithm. In section 3 we introduce the proposed model and the algorithm. Section 4 presents the theoretical analysis of the proposed model. Comparison of our algorithm to the best performing methods used in weakly supervised learning is given in section 5, based on the *WRENCH* framework (Zhang et al., 2021) which allows us to compare against other label models based on the *programmatic weak supervision (PWS)* paradigm (Ratner et al., 2016). Overall, we present a simple, flexible and yet the best performing label model for weakly supervised learning, based on strong mathematical foundations.

## 1.1 PROBLEM SETUP AND NOTATIONS

In this section, we will provide a concise overview of the problem setup and introduce the notations that will be utilized throughout the remainder of the paper. Vectors and matrices are shown in bold. All sets are denoted in calligraph font. $\mathbf{1}$ indicates the 1-vector for which all its entries are 1. $||\mathbf{x}||$ denotes the Euclidean norm of a vector $\mathbf{x}$. Given two sets $\mathcal{X}$ and $\mathcal{Y}$, $\mathcal{X} \backslash \mathcal{Y}$ denotes the set difference of $\mathcal{X}$ and $\mathcal{Y}$. Given a set of vectors $\mathcal{S} = \{\mathbf{a}_1, \mathbf{a}_2, ..., \mathbf{a}_r\}$ in $\mathcal{R}^m$, the convex hull of $\mathcal{S}$, denoted by $\text{Conv}(\mathcal{S})$, is the set of all convex combinations of vectors in $\mathcal{S}$. $\text{Col}(\mathbf{A})$ denotes the column space of $\mathbf{A}$. In weakly supervised learning, as noted in the introduction, the main goal is to return the synthetic label $\widetilde{\mathbf{y}}$ which is an estimate of $\mathbf{y}$, given the weak signals matrix $\mathbf{W} \in [0, 1]^{m \times nk}$. The $(k-1)n + i$-th entry of $\widetilde{\mathbf{y}}$ indicates the probability of $i$-th data point belonging to class $k$, hence $\widetilde{\mathbf{y}} \in [0, 1]^{nk}$. Let the unlabeled data points be denoted as $\mathbf{X} = [\mathbf{x}_1, ..., \mathbf{x}_n]$ and let $\mathbf{w_1}, ..., \mathbf{w_m}$ denote individual weak signals. From this point onwards, the parameter $m$ will represent the number of weak signals, $n$ will indicate the number of data points, and $k$ will denote the number of classes.

Table 1 is an illustration of weak signals along with the true label. Note that $\mathbf{y}$ is the ground-truth label which is not given, and the objective of label models is to return the synthetic label $\widetilde{\mathbf{y}}$ which is an estimate of $\mathbf{y}$.

Table 1: Illustration of weak signals and labels. Each label vector of length $nk$ gives information about $n$ data points that are classified into $k$ classes. In this case, there are 3 weak signals that gives information about 5 data points, where each data points are classified into 3 classes. The ground truth label $\mathbf{y}^\top$ indicates for each data point if it is in the corresponding class. When the weak signal does not give any information (i.e., when abstaining), we assign $\frac{1}{k}$ to the corresponding $\emptyset$ entry.

|  | Class 1 | | | | | Class2 | | | | | Class3 | | | | |
|---|---|---|---|---|---|---|---|---|---|---|---|---|---|---|---|
| $\mathbf{w}_1$: | 0.8 | $\emptyset$ | 0.0 | 0.8 | 0.4 | $\emptyset$ | 0.7 | $\emptyset$ | 0.2 | $\emptyset$ | $\emptyset$ | $\emptyset$ | 0.6 | $\emptyset$ | $\emptyset$ |
| $\mathbf{w}_2$: | 0.7 | 0.2 | $\emptyset$ | 0.6 | 0.3 | $\emptyset$ | $\emptyset$ | $\emptyset$ | $\emptyset$ | $\emptyset$ | $\emptyset$ | $\emptyset$ | $\emptyset$ | 0.3 | $\emptyset$ |
| $\mathbf{w}_3$: | $\emptyset$ | $\emptyset$ | $\emptyset$ | $\emptyset$ | $\emptyset$ | 0.4 | $\emptyset$ | $\emptyset$ | 0.4 | 0.6 | $\emptyset$ | $\emptyset$ | $\emptyset$ | $\emptyset$ | 0.9 |
| Data: | $x_1$ | $x_2$ | $x_3$ | $x_4$ | $x_5$ | $x_1$ | $x_2$ | $x_3$ | $x_4$ | $x_5$ | $x_1$ | $x_2$ | $x_3$ | $x_4$ | $x_5$ |
| $\mathbf{y}^\top$ | 1 | 0 | 0 | 1 | 0 | 0 | 1 | 0 | 0 | 0 | 0 | 0 | 1 | 0 | 1 |

## 2 RELEVANT WORK

Weakly supervised learning has been applied to variety of tasks, including computer vision (Chen & Batmanghelich, 2020), text classification (Chen & Batmanghelich, 2020) and sentiment analysis (Medlock & Briscoe, 2007). Weakly supervised learning is also studied in close relation to other branches of learning as well, including unsupervised learning (Chen & Batmanghelich, 2020), which does not require labeled input data, and self-supervised learning (Karamanolakis et al., 2021), that aims to extract information from the input data instead of relying on information from outside. Weak supervision algorithms output synthetically labeled datasets which are then used for training machine learning models such as transformer (Vaswani et al., 2017) and BERT (Devlin et al., 2019).

Majority of existing models make assumptions about the distribution of the data or the weak signals and utilize it to form a synthetic label for the dataset (Zhang et al., 2022). Ratner et al. (2016) uses a generative model, Fu et al. (2020) assumes certain degree of class balance, and Wu et al. (2023), Yu et al. (2022), Kuang et al. (2022) similarly utilize the assumptions to represent the distribution to formulate the synthetic label. Other methods like Arachie & Huang (2021) and Dawid & Skene (1979) assume a particular characteristic of the accuracy of the weak signals to produce the synthetic labels. Our method is philosophically similar to Constrained Label Learning (Arachie & Huang, 2021) in the sense that we define a feasible space for $\widetilde{\mathbf{y}}$. However, our method is fundamentally different to *Constrained Label Learning* (CLL) in that *RACH-Space* does not assume the prior knowledge of expected empirical rates of the weak signals.

Whilst *Hyper Label Model* (HLM) (Wu et al., 2022) does not need an ad hoc dataset-specific parameter, it considers the setup where the entries of weak signals are one of $\{0, 1, -1\}$. *RACH-Space* does not require the entries of weak signals to be an integer, and takes any input between $[0, 1]$ (for label models in PWS, it would be $[-1, 1]$), thus allowing weak signals to express its confidence for each data points in terms of probability.

The main optimization problem RACH-space solves is a minimization of the quadratic objective $f(\widetilde{\mathbf{y}}) = \widetilde{\mathbf{y}}^T \mathbf{A}^T \mathbf{A} \widetilde{\mathbf{y}} + \widetilde{\mathbf{y}}^T \mathbf{A}^T \widetilde{\mathbf{b}}$ for suitably defined $\mathbf{A}$ and $\widetilde{\mathbf{b}}$ over the unit simplex. In the setting we consider, the program in general admits infinitely many solutions. Given this fact, an arbitrary choice of an optimal solution of this program could lead to poor estimation of the underlying true label. RACH-space posits that there is a *safe region* (safe in terms of minimizing the error of the learned label) for $\frac{\widetilde{\mathbf{b}}}{n}$ to lie in. This region is within the convex hull of $\mathbf{A}$ and outside the second largest convex hull of $\mathbf{A}$. A pivotal aspect of RACH-space involves adjusting the pseudo-error rate to ensure that $\frac{\widetilde{\mathbf{b}}}{n}$ falls within the *safe region*. In our algorithm, these first two layers, the convex hull and the second largest convex hull, assume critical importance. We note that our approach is thematically related to the concept of convex layers or convex hull peeling, where a set of points are represented as a sequence of convex layers by progressively eliminating vertices from the convex hull (Calder & Smart, 2020; Dalal, 2004; Barnett, 1976; Chazelle, 1985).

## 3 PROPOSED MODEL AND ALGORITHM

In this section, we discuss our algorithm. In particular, we analyse the pseudo-error rate $\epsilon$ and propose a simple algorithm based on the bounds on $\epsilon$.

### 3.1 RACH-SPACE

Having established the expected empirical error rate in 1.2, we now aim to ascertain the bounds for the expected empirical rate of a random label assignment. First, we clarify what a label that does random classification of the data points might look like. Such a random signal could be $\mathbf{w} = \{1/k, 1/k, ...1/k\}$ where $k$ is the number of classes. In this case, the error rate of this weak signal would be $\frac{2}{k} - \frac{2}{k^2}$ by applying 1.2 on a random signal. Note that this is equivalent to $\frac{1}{2}$ for binary classification. We establish the bounds for $\epsilon$ in the following lemma.

**Lemma 3.1.** *If a weak signal is better than random labeling, the expected empirical of the weak signal is bounded by the following:*

$$0 \leq \epsilon \leq \frac{2}{k} - \frac{2}{k^2}, \tag{3.1}$$

*where $\epsilon$ is a pseudo-error rate, previously defined as the expected empirical error rate of a weak signal that is better than random classification of the data points.*

*Proof.* Consider any labeling vector $\mathbf{z} \in [0, 1]^{nk}$. The ground truth label has $n$ ones and $n(k-1)$ zeros. If we estimate the labels at random, the probability that a data point is in any given class is $\frac{1}{k}$. Thus, a false positive where $\mathbf{z}$ incorrectly labels a data point to be in certain class when it is not the case in ground-truth is $\mathbf{z}^\top (1 - \mathbf{y}) = \frac{1}{k} \times n(k-1)$. Similarly, a false negative where $\mathbf{z}$ incorrectly labels a data point to not be in a certain class when it should be according to the ground-truth is $(\mathbf{1} - \mathbf{z})^\top \mathbf{y} = \frac{k-1}{k} \times n$. Hence the expected empirical rate of this random labeling vector is: $\frac{1}{nk} \left( \mathbf{z}^\top (1 - \mathbf{y}) + (\mathbf{1} - \mathbf{z})^\top \mathbf{y} \right) = \frac{1}{nk} \left( \frac{1}{k} \times n(k-1) + \frac{k-1}{k} \times n \right) = \frac{2}{k} - \frac{2}{k^2}$. This is also the error

rate of the labeling vector $(\frac{1}{k}, ..., \frac{1}{k})$ independent of whether the data is class-imbalanced or not. The error rate is trivially bounded below by 0. $\qquad\square$

In the typical setup, $m \ll nk$. Given weak signals $\mathbf{w_1}, ..., \mathbf{w_m}$, we are interested in returning the synthetic label $\widetilde{\mathbf{y}}$ which is an optimal estimate of $\mathbf{y}$. Thus, we consider minimizing the error between the estimate and true label for unlabeled dataset $\mathbf{X} = [\mathbf{x}_1, ..., \mathbf{x}_n]$. With the definition of pseudo-error rate $\epsilon$, we have that $\epsilon$ has an upper bound of $\frac{2}{k} - \frac{2}{k^2}$ and a lower bound of 0, where the latter holds when the weak signal is the ground-truth label itself.

Our algorithm aims to capture the relation between the pseudo-error rate, weak signals and the overall data structure. To do so, we formulate a least squares problem for the synthetic label $\widetilde{\mathbf{y}}$ from 1.2, but instead of $\epsilon_i$ which we do not have knowledge of, we replace it with the pseudo-error rate $\epsilon$. Then, we introduce variables $\widetilde{\mathbf{b}} \in \mathcal{R}^m$ and $\mathbf{A} \in [0, 2]^{m \times nk}$. The functionality of $\mathbf{b}$ and $\mathbf{A}$ is to act as a bridge between weak signals and the synthetic label $\widetilde{\mathbf{y}}$, representing the cumulative effect of errors in weak signals. Let $\mathbf{A}$ be defined as $\mathbf{A} = 2\mathbf{W}$, where each row of $\mathbf{W}$ corresponds to the weak signals. Let $\widetilde{\mathbf{b}} \in \mathcal{R}^m$ with $\widetilde{b}_i = -nk \cdot \epsilon + \mathbf{w_i}^\top \mathbf{1} + n$. This depends on the pseudo-error rate $\epsilon$.

It is essential to note that while we can establish a bound for the pseudo-error rate $\epsilon$, the individual error rates of weak signals and other intricacies in the data structure remains unknown unless extra assumptions are made. Thus by defining the synthetic label $\widetilde{\mathbf{y}} \in [0, 1]^{nk}$ as the vector that satisfies $\mathbf{A}\widetilde{\mathbf{y}} = \widetilde{\mathbf{b}}$, RACH-Space solely hinges on this bound on the pseudo-error rate.

Our algorithm discussed in Section 3.2 adopts the most "conservative" choice of $\epsilon$ such that $\frac{\widetilde{\mathbf{b}}}{n}$ is in the *safe region*, which is explained in Section 4. The term "conservative" refers to *RACH-Space*'s strategy of maximizing the bounds for $\epsilon$, ensuring we account for the worst-case scenario. This reduces the risk of underestimating the error rate of each weak signals, thus providing a robust and resilient model. This is done by setting $\epsilon$ as its upper bound $\frac{2}{k} - \frac{2}{k^2}$ and decreasing it until $\frac{\widetilde{\mathbf{b}}}{n}$ is in the *safe region*.

Let $\mathbf{r}$ be the $\widetilde{\mathbf{b}}$ that Algorithm 1 chooses. This allows us to formulate the objective function as

$$\min_{\widetilde{\mathbf{y}} \in [0,1]^{nk}} \quad ||\mathbf{A}\widetilde{\mathbf{y}} - \mathbf{r}||^2 \quad \text{subject to} \quad \mathbf{1}^\top \widetilde{\mathbf{y}} = n. \tag{3.2}$$

## 3.2 ALGORITHM

---
**Algorithm 1** *RACH-Space*. See section 3.2 for details.

---
**Require:** $\alpha$ : Stepsize
**Require:** Weak signals $[\mathbf{w}_1, ..., \mathbf{w}_m]$
    $\mathbf{A} \leftarrow \mathbf{A} = 2\mathbf{W}$
    $\widetilde{\mathbf{b}} \leftarrow \widetilde{b}_i = -nk \cdot \left(\frac{2}{k} - \frac{2}{k^2}\right) + \mathbf{w}_i^\top \mathbf{1} + n$ (Initialize $\mathbf{b}$)
    $\widetilde{\mathbf{y}} \leftarrow \widetilde{\mathbf{y}} \sim U(0, 1)^n$ initialize $\widetilde{\mathbf{y}}$
    $\mathcal{H}_1$ is the set of columns that form the convex hull of $\mathbf{A}$
    $\mathcal{H}_2$ are the remaining columns, i.e., $\mathcal{H}_2 = \mathbf{A} \backslash \mathcal{H}_1$
    **while** $\frac{\widetilde{\mathbf{b}}}{n} \in \text{Conv}(\mathcal{H}_2)$ **do**
        $\widetilde{\mathbf{b}} \leftarrow \widetilde{b}_i = -nk \cdot \left(\frac{2}{k} - \frac{2}{k^2} - \alpha\right) + \mathbf{w_i}^\top \mathbf{1} + n$
    **end while**
    $\mathbf{r} = \widetilde{\mathbf{b}}$.
    Add a row of 1's to $\mathbf{A}$ and append $n$ at the bottom of $\mathbf{r}$
    **while** $\widetilde{\mathbf{y}}$ not converged **do**
        Update $\widetilde{\mathbf{y}}$ with gradient descent by optimizing $f(\widetilde{\mathbf{y}}) = ||\mathbf{A}\widetilde{\mathbf{y}} - \mathbf{r}||^2$
        Clip $\widetilde{\mathbf{y}}$ to $[0, 1]^{nk}$
    **end while**
    **return** $\widetilde{y}$

---

See Algorithm 1 for the pseudo-code of our proposed algorithm *RACH-Space*. $m$ is the number of weak signals, $n$ is the number of data points and $k$ is the number of classes. $\widetilde{\mathbf{y}} \in [0, 1]^{nk}$ is our synthetic label. We denote $\text{Conv}(\mathcal{H}_1)$ as the convex hull generated by the columns of $\mathbf{A}$ and

Conv($\mathcal{H}_2$) as the second largest convex hull strictly inside Conv($\mathcal{H}_1$), where $\mathcal{H}_1$ are the columns of $\mathbf{A}$ that defines Conv($\mathcal{H}_1$) and $\mathcal{H}_2$ are the remaining columns of $\mathbf{A}$. The algorithm updates $\widetilde{\mathbf{b}}$ using the given step size $\alpha$. As all the entries of $\mathbf{A}$ are in $[0, 2]$, this has the effect of pushing $\frac{\widetilde{\mathbf{b}}}{n}$ outside of Conv($\mathcal{H}_2$) as it positively increments all entries of $\widetilde{\mathbf{b}}$. If $\widetilde{\mathbf{b}}$ is already outside of Conv($\mathcal{H}_2$), then the algorithm sets $\widetilde{\mathbf{b}} = -nk\left(\frac{2}{k} - \frac{2}{k^2}\right) + \mathbf{w_i}^\top \mathbf{1} + n$. This can be understood as pushing $\widetilde{\mathbf{b}}$ into the *safe region*, which is described in Section 4. Finally, the algorithm uses gradient descent to optimize a modified least squares objective equation 3.2, where the modification adds the sum to one constraint, and finally applies clipping to satisfy the constraint that $\widetilde{\mathbf{y}} \in [0, 1]^{nk}$.

## 3.3 UPDATING $\frac{\widetilde{\mathbf{b}}}{n}$ OUT OF CONV($\mathcal{H}_2$)

A central part of *RACH-Space* is its decision of $\widetilde{\mathbf{b}}$ through the convex hull structure inherent in the column space of $\mathbf{A}$. As noted before, all the entries of $\mathbf{A}$ are in $[0, 2]$, as $\mathbf{A}$ is defined by $2\mathbf{W}$, where the entries of weak signal matrix $\mathbf{W}$ are in $[0, 1]$. Updating $\frac{\widetilde{\mathbf{b}}}{n}$ out of Conv($\mathcal{H}_2$) requires the computation of the column vectors $\mathcal{H}_1$ of $\mathbf{A}$ to identify the remaining columns $\mathcal{H}_2$ of $\mathbf{A}$ and checking whether $\frac{\widetilde{\mathbf{b}}}{n}$ is in Conv($\mathcal{H}_2$) each time $\widetilde{\mathbf{b}}$ is updated. To compute $\mathcal{H}_1$, we use Qhull (Barber et al., 1996). This can be expensive in practice when the dimension of the columns of $\mathbf{A}$ is high. In our case, as $\mathbf{A}$ has $nk$ columns of dimension $m$, the time complexity to compute $\mathcal{H}_1$ is $O((nk)^{\lfloor \frac{m}{2} \rfloor})$ (Barber et al., 1996). This is $O(n)$ for $m = 2, 3$ and $O(n^2)$ for $m = 4, 5$. During experiments, to make the run time reasonable, we reduced the number of weak signals by dividing them into five chunks in given order and getting the average of each chunks. Reducing the number of weak signals this way did not have negative impact for the performance of *RACH-Space*, as there was little difference in performance when seven chunks were averaged instead. In addition, with the reduced number of weak signals, *RACH-Space* still had the best overall performance on 14 *WRENCH* benchmark datasets for weak supervision including 4 state-of-art performance (Zhang et al., 2021) compared to 7 other existing label models. *RACH-Space* is not restricted to this method of reducing the number of weak signals, and one can aggregate the weak signals using a method of their choice, such as clustering. In our numerical experiments, *RACH-Space* is not sensitive to the particular method of aggregation in practice. Checking whether $\frac{\widetilde{\mathbf{b}}}{n}$ is in Conv($\mathcal{H}_2$) can be done readily by certifying whether the associated linear program is feasible.

## 4 SAFE REGION

Section 3 introduced the concept of *safe region*. This region ensures that $\widetilde{\mathbf{y}}$, the synthetic label computed by *RACH-Space*, does not overlook the parts of weak signals in $\mathbf{W}$ with the strongest class indicators. In this section, we are going to show that *safe region* of $\frac{\widetilde{\mathbf{b}}}{n}$ is the area inside Conv($\mathcal{H}_1$) but outside Conv($\mathcal{H}_2$). Since $\mathbf{A}$ is defined as $2\mathbf{W}$, larger entries in $\mathbf{A}$ correspond to stronger class indication in $\mathbf{W}$. As all entries in $\mathbf{A}$ are in $[0, 2]$, the extreme points of the set of columns in $\mathbf{A}$ correspond to parts of the weak signal $\mathbf{W}$ with the strongest class indication. In practice, the number of columns in matrix $\mathbf{A}$ (denoted as $nk$) is considerably greater than the column dimension ($m$). Consequently, there exists a suitable subset of these columns that define the convex hull.

The following theorem demonstrates that if $\frac{\widetilde{\mathbf{b}}}{n} \in$ Conv(Col($\mathbf{A}$)), the least squares problem admits infinitely many solutions.

$$\min_{\widetilde{\mathbf{y}} \geq \mathbf{0}} \quad \left\| \mathbf{A}\widetilde{\mathbf{y}} - \widetilde{\mathbf{b}} \right\|^2 \quad \text{subject to} \quad \mathbf{1}^T \widetilde{\mathbf{y}} = n, \tag{4.1}$$

where $\{\widetilde{\mathbf{y}} | \widetilde{\mathbf{y}} \in [0, 1], \widetilde{\mathbf{y}} \geq \mathbf{0}\}$ has been replaced with the set of vectors with non-negative entries. The proof of the theorem relies on the notion of affine independence which we define below.

**Definition 4.1.** *The vectors $\mathbf{p}_1, \mathbf{p}_2, ..., \mathbf{p}_k$ are affinely independent if the vectors $\mathbf{p}_2 - \mathbf{p}_1, \mathbf{p}_3 - \mathbf{p}_1, ..., \mathbf{p}_k - \mathbf{p}_1$ are linearly independent. Otherwise, the vectors are affine dependent.*

**Theorem 4.2.** *For $\frac{\widetilde{\mathbf{b}}}{n} \in$ Conv(Col($\mathbf{A}$)) and $nk > m + 1$, the optimization program in 4.1 has infinitely many solutions.*

*Proof.* This proof is similar to the standard proof used to establish Caratheodory's theorem in convex analysis. Consider the vectors $\mathbf{a}_2 - \mathbf{a}_1, \mathbf{a}_3 - \mathbf{a}_1, ..., \mathbf{a}_{nk} - \mathbf{a}_1$ where $\mathbf{a}_i$ denotes the i-th column of $\mathbf{A}$. Since $nk > m + 1$, these set of vectors are affinely dependent. This implies that there exists constants $d_2, d_3, .., d_{nk}$, which are all not zero, such that $\sum_{i=2}^{nk} d_i(\mathbf{a}_i - \mathbf{a}_1) = \mathbf{0}$. Equivalently, this can be written as $\sum_{i=1}^{nk} d_i \mathbf{a}_i = \mathbf{0}$ where $d_1 = -\sum_{i=2}^{nk} d_i$. For latter use, note that $\sum_{i=1}^{nk} d_i = 0$. Using the fact that $\frac{\widetilde{\mathbf{b}}}{n} \in \text{Conv}(\text{Col}(\mathbf{A}))$, there exists a $\widetilde{\mathbf{y}} \in [0, 1]^{nk}$ with $\mathbf{1}^T \widetilde{\mathbf{y}} = n$ such that $\mathbf{A}\widetilde{\mathbf{y}} = \widetilde{\mathbf{b}}$. We now propose a new representation of $\widetilde{\mathbf{b}}$ as follows: $\widetilde{\mathbf{b}} = \sum_{i=1}^{nk} (\widetilde{y}_i + \alpha d_i) \mathbf{a}_i$ where $\alpha > 0$. For this to lead to another optimal solution, it suffices to check that $\widetilde{y}_i + \alpha d_i \geq 0$ for all values of $i$. For $d_i \geq 0$, $\widetilde{y}_i + \alpha d_i \geq 0$ holds automatically. The relevant cases are the values of $i$ for which $d_i < 0$. Setting $\alpha^* = \min\limits_{i:d_i<0} -\frac{\widetilde{y}_i}{d_i}$, we see that $\widetilde{y}_i + \alpha^* d_i \geq 0$ and $\sum_{i=1}^{nk}(\widetilde{y}_i + \alpha^* d_i) = n$. Therefore, we have a new solution $\widetilde{\mathbf{z}}$ which differs from $\widetilde{\mathbf{y}}$. Once we have this pair of solutions, we can generate infinitely many solutions using $\widetilde{\mathbf{w}} = \lambda\widetilde{\mathbf{y}} + (1 - \lambda)\widetilde{\mathbf{z}}$ for any $\lambda \in (0, 1)$. $\qquad\square$

## 4.1 *Safe region* WITHIN CONV($\mathcal{H}_1$)

In this section, we show that the *safe region* of $\frac{\widetilde{\mathbf{b}}}{n}$ is further restricted inside $\text{Conv}(\mathcal{H}_1)$. In particular, we articulate why we have that $\frac{\widetilde{\mathbf{b}}}{n} \notin \text{Conv}(\mathcal{H}_2)$.

**Lemma 4.3.** *If $\frac{\widetilde{\mathbf{b}}}{n} \in Conv(\mathcal{H}_2)$, the computed synthetic label can converge to a label where all the entries corresponding to the the extreme points of Conv(Col($\mathbf{A}$)) are labeled $0$.*

*Proof.* Suppose $\frac{\widetilde{\mathbf{b}}}{n} \in \text{Conv}(\mathcal{H}_2)$. Then, $\frac{\widetilde{\mathbf{b}}}{n}$ has a convex combination of columns in $\mathcal{H}_2$. Denote the coefficients arising from this convex combination as $\widetilde{\mathbf{y}}$. By applying **Thm 4.2** for $\frac{\widetilde{\mathbf{b}}}{n} \in \text{Conv}(\mathcal{H}_2)$, there exist infinitely many solutions for $\widetilde{\mathbf{y}}$ as well. By construction, solving $\mathbf{A}\widetilde{\mathbf{y}} = \widetilde{\mathbf{b}}$ subject to $\widetilde{\mathbf{y}} \in [0, 1]^{nk}$ and $\mathbf{1}^\top\widetilde{\mathbf{y}} = n$ can converge to a synthetic label $\widetilde{\mathbf{y}}$ where all entries corresponding to the columns of $\mathcal{H}_1$ are labeled $0$. $\qquad\square$

**Remark 4.4.** *If $\frac{\widetilde{\mathbf{b}}}{n} \notin Conv(\mathcal{H}_2)$, the computed synthetic label cannot converge to a label where all the entries corresponding to the the extreme points of Conv(Col($\mathbf{A}$)) are labeled $0$.*

**Lemma 4.3** and **Remark 4.4** together show that updating $\frac{\widetilde{\mathbf{b}}}{n}$ out of $\text{Conv}(\mathcal{H}_2)$ prevents the synthetic label from incorrectly converging to $0$ at the extreme points of $\text{Conv}(\text{Col}(\mathbf{A}))$. Note, the extreme points of $\text{Conv}(\text{Col}(\mathbf{A}))$ correspond to parts of the weak signal $\mathbf{W}$ with the strongest class indication.

This is done in our algorithm by decreasing $\epsilon$ via step size $\alpha$. As previously mentioned in Section 3.3, decreasing $\epsilon$ via step size $\alpha$ has the effect of increasing each entry of $\widetilde{\mathbf{b}}$, and as columns in $\mathbf{A}$ are non negative with entries in $[0, 2]$ this slowly pushes $\frac{\widetilde{\mathbf{b}}}{n}$ out of $\text{Conv}(\mathcal{H}_2)$. Taking into account that strong class indications of weak signals are not necessarily correct indications as weak signals can be arbitrarily erroneous, this is not a hard guarantee of increased accuracy of the synthetic label. However, this process effectively selects an $\epsilon$ that minimizes the likelihood of the synthetic label converging arbitrarily.

By decreasing $\epsilon$ until $\frac{\widetilde{\mathbf{b}}}{n}$ is pushed out of $\text{Conv}(\mathcal{H}_2)$, *RACH-Space* selects the most conservative choice of $\epsilon$ that can systematically avoid the case where the strongest class indications coming from the extreme points of $\text{Conv}(\text{Col}(\mathbf{A}))$ are ignored. This is supported by our experiments in Section 5 on empirical data, which supports the claim that having $\frac{\widetilde{\mathbf{b}}}{n}$ inside $\text{Conv}(\mathcal{H}_2)$ makes the algorithm prone to arbitrary convergence, and that once such conservative choice of $\widetilde{\mathbf{b}}$ is made, *RACH-Space* shows state of art performance compared to all other existing label models.

## 5 EXPERIMENTS

We evaluate our proposed method on the *WRENCH* weak supervision benchmark (Zhang et al., 2021). The datasets in the *WRENCH* benchmark are in accordance with the *Programmatic weak supervision (PWS)* (Ratner et al., 2016). In *PWS*, labeling functions (LFs) process data points and

Table 2: 14 classification datasets from the weak supervision benchmark  (Zhang et al., 2021)

| Dataset | Census | Yelp | Youtube | CDR | Basketball | AGNews | TREC | SemEval | ChemProt | Spouse | Imdb | Commercial | Tennis | SMS |
|---|---|---|---|---|---|---|---|---|---|---|---|---|---|---|
| Task | income | sentiment | spam | bio relation | video frame | topic | question | relation | chemical relation | relation | sentiment | video frame | video frame | spam |
| #class | 2 | 2 | 2 | 2 | 2 | 4 | 6 | 9 | 10 | 2 | 2 | 2 | 2 | 2 |
| metric | F1 | acc | acc | F1 | F1 | acc | acc | acc | acc | F1 | acc | F1 | F1 | F1 |
| #LF | 83 | 8 | 10 | 33 | 4 | 9 | 68 | 164 | 26 | 9 | 8 | 4 | 6 | 73 |
| #train | 10083 | 30400 | 1586 | 8430 | 17970 | 96000 | 4965 | 1749 | 12861 | 22254 | 20000 | 64130 | 6959 | 4571 |
| #validation | 5561 | 3800 | 120 | 920 | 1064 | 12000 | 500 | 178 | 1607 | 2801 | 2500 | 9479 | 746 | 500 |
| #test | 16281 | 3800 | 250 | 4673 | 1222 | 12000 | 500 | 600 | 1607 | 2701 | 2500 | 7496 | 1098 | 500 |

output a noisy label. Thus, LFs act as a form of weak supervision and are considered weak signals. All LFs in *WRENCH* benchmark are from the original authors of each dataset  (Zhang et al., 2021). The LFs yield weak signals with entries in $\{-1, 0, +1\}$, where $+1$ and $0$ signify class membership, and $-1$ indicates abstention. *RACH-Space* convert this into weak signal format we use, where each weak signals now have entries in $\{\emptyset, 0, 1\}$ where $\emptyset$ indicates abstention. We highlight that this is a slightly modified setup compared to the setup for *RACH-Space* where the weak signals can take any values in $\{\emptyset, [0, 1]\}$, allowing room for weak signals to indicate class membership in terms of probability. For the sake of comparison based on the *WRENCH* benchmark, we apply this setting where the entries of weak signals are in $\{\emptyset, 0, 1\}$. Our theoretical analysis regarding the convex hulls still hold with this assumption on the entries of weak signals.

## 5.1    EXPERIMENT: PERFORMANCE OF LABEL MODELS

Our empirical experiments were conducted using the metrics provided by the benchmark  (Zhang et al., 2021) for each dataset, where each metric value is averaged over 5 runs. Our experiments are conducted on 14 datasets on *WRENCH* benchmark, which covers various classification tasks and includes multi-class classification. Table 2 shows the statistics of each dataset.

**Label models:** (1) *Majority voting* (*MV*). For each point, the predicted label is determined by choosing the most common label provided by the $m$ weak signals. Formally, it would simply count the number of 1's and 0's in the $m$ weak signals for the corresponding data point and choose whichever is more common as the data point's predicted label. (2) *Dawid-Skene* (*DS*)  (Dawid & Skene, 1979) assumes a naive Bayes distribution over the weak signals and the ground-truth label to estimate the accuracy of each weak signals. (3) *Data Programming* (*DP*)  (Ratner et al., 2016) describes the distribution of $p(L, Y)$ as a factor graph, where $L$ is the LF and $Y$ is the ground-truth label. (4) *MeTaL*  (Ratner et al., 2019) models $p(L, Y)$ via Markov Network, and  Ratner et al. (2018) uses it for weak supervision. (5) *FlyingSquid* (*FS*)  (Fu et al., 2020) models $p(L, Y)$ as a binary Ising model and requires label prior. It is designed for binary classification but one-versus-all reduction method was applied for multi-class classification tasks. (6) *Constrained Label Learning* (*CLL*)  (Arachie & Huang, 2021) requires prior knowledge of the expected empirical rates for each weak signals to compute a constrained space from which they randomly select the synthetic labels from. For our experiments, we ran *CLL* with the assumption that all weak labels are better than random. (7) *Hyper Label Model* (*HLM*)  (Wu et al., 2022) trains the model on synthetically generated data instead of actual datasets. Note that the difference in our experiment results from  Wu et al. (2022) is because their experiments were conducted in transductive setting  (Mazzetto et al., 2021), where data points used in learning is also used to evaluate the learned model. Hence their experiments are done where the train, validation and test datasets are merged for the label models to learn and to be evaluated. Our experiments are done in the original setup on *WRENCH*  (Zhang et al., 2021) where the label models are trained on train data and evaluated on test data.

**Results:**  *RACH-Space* outperforms all other existing models, including the previous best-performing label model *HLM* by $0.72$ points. For the experiments, *RACH-Space* reduced the number of weak signals to $5$ by simply averaging five chunks of weak signals in given order, and a step size $\alpha = 0.01$ was chosen whilst taking $\frac{\widetilde{\mathbf{b}}}{n}$ out of $\text{Conv}(\mathcal{H}_2)$. The number of weak signals did not have significant impact on the performance of *RACH-Space*, as the results were similar as long as there were more than 1 weak signal, in which case there would not be a convex hull. Still, there were some exceptions for the parameters used. As for the Imdb dataset, the number of weak signals was reduced even smaller to $3$ and as for the SMS dataset, step size $\alpha = 0.001$ was chosen because the step size needed to be smaller than $\alpha = 0.01$ in order to take $\frac{\widetilde{\mathbf{b}}}{n}$ into the *safe region*. Our algorithm

Table 3: Label model performance

| Dataset | Census | Yelp | Youtube | CDR | Basketball | AGNews | TREC | SemEval | ChemProt | Spouse | Imdb | Commercial | Tennis | SMS | AVG. |
|---|---|---|---|---|---|---|---|---|---|---|---|---|---|---|---|
| MV | 32.80 | 70.21 | 84.00 | 60.31 | 16.33 | 63.84 | 60.80 | 77.33 | **49.04** | 20.80 | **71.04** | 85.28 | 81.00 | **23.97** | 56.91 |
| DS | 47.16 | **71.45** | 83.20 | 50.43 | 13.79 | 62.76 | 50.00 | 71.00 | 37.59 | 15.53 | 70.60 | **88.24** | 80.65 | 4.94 | 53.38 |
| DP | 12.60 | 69.38 | 82.24 | 55.12 | 17.39 | **63.95** | 63.28 | 71.00 | 46.86 | 21.21 | 70.96 | 77.28 | **83.14** | **23.96** | 54.16 |
| MeTaL | 38.12 | 55.29 | 60.40 | 30.80 | 0 | 64.18 | 50.48 | 54.17 | **49.84** | 20.13 | 69.96 | 77.95 | 80.54 | 23.83 | 48.26 |
| FS | 17.77 | **71.68** | 78.40 | **63.22** | 17.39 | 63.55 | 57.80 | 12.50 | 46.55 | 21.14 | 70.36 | 81.84 | 77.79 | 23.86 | 50.28 |
| CLL | 26.99 | 51.89 | 57.60 | 24.97 | 16.12 | **64.83** | 61.24 | 78.83 | 46.79 | **22.56** | 49.52 | 38.32 | 25.47 | 15.34 | 41.46 |
| HLM | **56.30** | 69.40 | **85.60** | 60.60 | **17.60** | 63.70 | **66.20** | **82.50** | 46.73 | 20.82 | **71.84** | 82.76 | **82.44** | 23.19 | **59.27** |
| RACH-Space | **52.82** | 67.37 | **87.60** | **70.04** | **17.66** | 62.93 | **65.80** | **81.16** | 44.99 | **44.51** | 69.60 | 78.44 | 73.28 | 23.63 | **59.99** |

Table 4: Comparison between $\frac{\widetilde{\mathbf{b}}}{n} \in \mathrm{Conv}(\mathcal{H}_2)$ and $\frac{\widetilde{\mathbf{b}}}{n} \notin \mathrm{Conv}(\mathcal{H}_2)$ (*safe region*)

| Dataset | Census | Yelp | Youtube | CDR | Basketball | AGNews | TREC | SemEval | ChemProt | Spouse | Imdb | Commercial | Tennis | SMS |
|---|---|---|---|---|---|---|---|---|---|---|---|---|---|---|
| $\frac{\widetilde{\mathbf{b}}}{n} \in \mathrm{Conv}(\mathcal{H}_2)$ | 38.22 | 64.47 | 82.80 | 69.55 | 17.66 | 10.39 | 12.16 | 2.90 | 5.63 | 44.51 | 37.44 | 66.11 | 36.27 | 23.36 |
| $\frac{\widetilde{\mathbf{b}}}{n} \notin \mathrm{Conv}(\mathcal{H}_2)$ | 52.82 | 67.37 | 87.60 | 70.04 | 17.66 | 62.93 | 65.80 | 81.16 | 44.99 | 44.51 | 69.60 | 78.44 | 73.28 | 23.63 |

includes the verification step of checking $\frac{\widetilde{\mathbf{b}}}{n} \in \mathrm{Conv}(\mathcal{H}_1)$ and $\frac{\widetilde{\mathbf{b}}}{n} \notin \mathrm{Conv}(\mathcal{H}_2)$ before solving the objective function, and it was verified for each empirical dataset during the experiment. During the reduction of weak signals when there are more than $5$ weak signals for a dataset, the entries are no longer integers, and take values in $[0, 1]$. Since *RACH-Space* does not assume that an entry in a weak signal takes an integer value, this is not a problem for *RACH-Space*. This is also why we do not conduct additional experiments on datasets outside of *WRENCH* framework on datasets where the weak signals can have fractional inputs. Our experimental results closely align with the results in Zhang et al. (2021), and is easily reproducible by using the datasets in the *WRENCH* benchmark. In our experiment we included all label models that showed the best performance for at least one dataset in the *WRENCH* benchmark.

## 5.2 EXPERIMENT: TESTING THE ACCURACY OUTSIDE THE *safe region*

We also empirically evaluate the effect of moving $\frac{\widetilde{\mathbf{b}}}{n}$ into the *safe region*. We use the same setup for *RACH-Space* but instead push $\frac{\widetilde{\mathbf{b}}}{n}$ to be inside $\mathrm{Conv}(\mathcal{H}_2)$, i.e. outside of the *safe region* and compare the results. We used the same method of updating $\widetilde{\mathbf{b}}$, but in the pushing it in the opposite direction using a negative step size of same size. Results are summarized in Table 4, and the empirical data confirms our analysis of *safe region*.

## 6 CONCLUSION

In the present work we propose a novel algorithm, *RACH-Space*, for classification in ensemble learning setting. In particular, we illustrate its applications in weakly supervised learning. We analyze the geometric structure hidden in the space related to weak signals. In particular we identify a convex hull structure that arises from a generic set of weak signals. We apply our analysis to make a conservative yet judicious selection for the pseudo-error rate. Our method performs competitively to all existing label models on commonly used weak supervision benchmarks which spans various classification tasks on real world datasets. *RACH-Space* not only outperforms other models on average over $14$ benchmark real world datasets, but also significantly enhance the state of art results in $4$ of them. Overall, in weakly supervised scenarios, *RACH-Space* proves to be both straightforward and robust, consistently delivering competitive performance on a wide range of classification tasks. *RACH-Space*'s most promising quality lies in its simplicity, from which we hope to replace traditional methods such as majority voting for extended problems outside of weak supervision.

## REPRODUCIBILITY STATEMENT

For reproducibility of the empirical results, we include our code *RACH-Space* in the supplementary material. This material contains instructions for installing the WRENCH project needed to load the datasets and weak labels and running the RACH-Space code.

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
