# OpenReview forum: "RACH-Space: Reconstructing Adaptive Convex Hull Space with applications in weak supervision"
_ICLR.cc/2024/Conference — Submitted to ICLR 2024_

### Official Review · Reviewer_16fQ · 2023-10-23

**Soundness:** 1 poor
**Presentation:** 1 poor
**Contribution:** 2 fair
**Rating:** 3
**Confidence:** 2

**Summary:**

The authors proposed a novel convex-hull-based method for weakly supervised learning. The proposed method is simple and works under a minimal assumption.

**Strengths:**

1. The proposed method is novel, as far as I know.
1. The proposed method outperformed other existing methods on average in the numerical experiments.

**Weaknesses:**

Overall, the presentation of the paper needs essential improvement in the problem-setting explanation, including the introduction section.
1. The definition of "weakly supervised learning" in this paper is not clear, which makes it challenging to understand the motivation of the paper and which applications it might have. The authors referred to some papers in the first and second paragraphs in the introduction section (Shin et al., 2015), (Mintz et al., 2009),  (Chen & Batmanghelich, 2020), and (Karamanolakis et al., 2021). Also, the authored mentioned (Ratner et al., 2016). Although those papers use the phrase "weakly supervised learning," the specific problem settings that they are tackling are different from the authors' formulation in Section 1.1. Hence, the authors are more responsible than usual to state the problem setting in a self-contained and rigorous manner, so that readers can understand which symbols correspond to which data in real applications and what types of assumptions are imposed.
1. The authors do not mention the link between $\\mathbf{W}$ and $\\mathbf{y}$. If there is no clear link between $\\mathbf{W}$ and $\\mathbf{y}$, we do not need to call the problem setting "weakly supervised learning" since $\\mathbf{W}$ is no longer a supervision.
1. As above, since the problem setting itself is unclear, readers cannot understand the importance of the area and this paper or discuss whether the proposed method is reasonable or not. I encourage the authors to make the assumptions and motivations clearer, ideally in the introduction section.
1. The RACH-space's main idea is not written in the Introduction section, which significantly decreases the readability. I strongly encourage the authors to explain in the introduction section what is the main idea of the proposed method and why the idea solves the issues of existing methods.
1. The definition of the *Majority Voting* is unclear.

**Questions:**

1. In which application cases, your problem setting in Section 1.1 is applicable? Could you provide specific examples? Specifically, in which case do we have multiple weak supervising signals?
1. The expected error is defined as $\\mathbf{E} [\\mathbf{w}\_\\mathbf{i} - \\mathbf{y}]$. I assume $\\mathbf{E}$ indicates the expectation operator, but what is the random variable of which we consider the expectation here? Also, you are interested in the difference between $\\mathbf{w}\_\\mathbf{i}$ and $\\mathbf{y}$. Does it mean that you assume the value of $\\mathbf{w}\_\\mathbf{i}$ is close to $\\mathbf{y}$? I do not see such assumptions in the paper.
1. If you assume the value of $\\mathbf{w}\_\\mathbf{i}$ is close to $\\mathbf{y}$ and you are interested in the error between them, why can the error be negative? Don't you need to consider the squared or absolute error or the cross entropy, which is a more natural choice for one-hot vector prediction?
1. Related to that, you mentioned "Note, Majority Voting and RACH-Space assume that, on average, the weak labels are better than random." Here, what does *better* mean? Could you mathematically define it?
1. Also, if you assume the value of $\\mathbf{w}\_\\mathbf{i}$ is an estimate of $\\mathbf{y}$, why the range of each entry of $\\mathbf{W}$ is $\\mathcal{R}$, which implies it can take any real value outside of $[0, 1]$? Is it a typo?
1. You mentioned many times *Majority Voting* but you do not define *Majority Voting*. What is that? Also, why do you not use *Majority Voting*? Why don't you compare your proposed method with the *Majority Voting*?

---

> ### Author Response · Authors · 2023-11-15
>
> We appreciate the feedback from the reviewer. Based on the feedback, we have now re-written the introduction. In the revised version, we state the problem in a self-contained manner. Further, the link between $\mathbf{W}$ and $\mathbf{y}$ is now shown in equations 1.1 and 1.2. We also now discuss the main idea of *RACH-Space* in the introduction and highlight the contributions.
>
> >Q1. In which application cases, your problem setting in Section 1.1 is applicable? Could you provide specific examples? Specifically, in which case do we have multiple weak supervising signals?
>
> There are numerous real world scenarios where the problem setting in Section 1.1 is applicable, and all of the $14$ real world datasets are from the public weak supervision benchmark WRENCH. For example, each weak signal can come from different sources of information, or/and be obtained at different timings and environments. Denoted by *#LFs* in Table 2 of our paper, real world scenarios involve different numbers of weak signals.
>
> >Q2. The expected error is defined as $\mathbf{E[w_i-y]}$. I assume $\mathbf{E}$ indicates the expectation operator, but what is the random variable of which we consider the expectation here? Also, you are interested in the difference between $\mathbf{w_i}$ and $\mathbf{y}$. Does it mean that you assume the value of $\mathbf{w_i}$ is close to $\mathbf{y}$? I do not see such assumptions in the paper.
>
> We appreciate the comment. This was part of our previous version that caused some confusion about the expected empirical error rate, so we modified the draft to clarify the definition of the expected empirical error rate. Each row of $\mathbf{W}$ represents a weak signal that gives probabilities for data points being in a certain class. So yes, the rows of $\mathbf{W}$, $\mathbf{w_i}$, are weak signals, which by definition are incomplete and noisy representations of the label vector $\mathbf{y}$.
>
> >Q3. If you assume the value of $\mathbf{w_i}$ is close to $\mathbf{y}$ and you are interested in the error between them, why can the error be negative? Don't you need to consider the squared or absolute error or the cross entropy, which is a more natural choice for one-hot vector prediction?
>
> Thank you for mentioning this. We clarified the definition of expected empirical error rate, for which the error cannot be negative (see 1.1 and 1.2). The expected empirical error rate is based on the sum of false negatives and false positives, which is one metric for weakly supervised learning.
>
> >Q4. Related to that, you mentioned "Note, Majority Voting and RACH-Space assume that, on average, the weak labels are better than random." Here, what does better mean? Could you mathematically define it?
>
> If one uses random labeling i.e., predict the label of a data point using a uniform probability of $\frac{1}{k}$, the expected empirical error rate is $\frac{2}{k}-\frac{2}{k^2}$. Better in this context means that the average expected empirical error rate of a weak signal is lower than $\frac{2}{k}-\frac{2}{k^2}$. In the revised version, we clarify that we do not assume prior knowledge nor a set bound for the average empirical error rates of weak signals. Instead, we update our parameter, now called pseudo-error rate, starting from $\frac{2}{k}-\frac{2}{k^2}$, which is the error rate of the weak signal that simply does random classification of the data points.
>
> >Q5. Also, if you assume the value of $\mathbf{w_i}$ is an estimate of $\mathbf{y}$, why the range of each entry of $\mathbf{W}$ is $\mathcal{R}$, which implies it can take any real value outside of $[0,1]$? Is it a typo?
>
> We appreciate the comment. This is a typo and has been corrected in the revised version.
>
> >Q6. You mentioned many times *Majority Voting* but you do not define *Majority Voting*. What is that? Also, why do you not use *Majority Voting*? Why don't you compare your proposed method with the *Majority Voting*?
>
> In the revised version, we have included a more detailed definition of *Majority Voting*. We did use *Majority Voting* in our numerical experiments. The first method in Table 3 is *Majority Voting*.

---

> ### Comment · Reviewer_16fQ · 2023-11-23
>
> Thank you for your comments and the revision. They partially solved my concerns, but it involved too much revisions. While it might have a potential for future publication, I believe the current draft is not ready for publication. This is why we keep the score.
>
> Further comments which might help your paper's further improvement:
>
> - Regarding Q1, just mentioning benchmark datasets is not persuasive. Your paper's readers would want to see an explanation about when in real applications we do not have ground truth labels but multiple weak signals and why.
>
> - Regarding Q6, I would suggest that the author explain why majority voting is not always good. This must be one starting point of your work. Also, why do we not consider simply averaging the weak signals?

---

### Official Review · Reviewer_LMCF · 2023-10-24

**Soundness:** 2 fair
**Presentation:** 2 fair
**Contribution:** 3 good
**Rating:** 5
**Confidence:** 4

**Summary:**

This paper proposes a classification method for ensemble learning and proves its effectiveness in weakly supervised learning. Specifically, given the data points and the weak signal matrix, this paper proposes a novel method to compute the label vector $\widetilde{\mathbf{y}}$ which approximates the ground truth $\mathbf{y}$. Then the problem is transferred to solving an optimization instance and finding the solution in a convex hull.

**Strengths:**

(1) The proposed optimization method for finding an approximate label vector is kind of novel.
(2) The framework for weakly supervised learning connects machine learning and high-dimensional geometry.
(3) The experimental results indicate the effectiveness of the proposed algorithm.

**Weaknesses:**

(1) The background is not very clear, especially for the weak signal matrix $\mathbf{W}$. For example, where does the matrix $\mathbf{W}$ come from?
(2) Some preliminaries are missing. Why the expected error rate $\epsilon_i$ is equal to $\mathbb{E}[\mathbf{w}_i - \mathbf{y}]$? Does each row of $\mathbf{W}$ kind of represent the label vector $\mathbf{y}$?
(3) As stated in this paper, the computation of $\mathcal{H}_1$ takes time $O((nk)^{\lfloor \frac{m}{2} \rfloor})$ which is prohibitive in practice. The author divided $\mathbf{W}$ into several parts and took their average to reduce the value of $m$. Although the author claimed that this does not make a negative impact on the performance, it cannot convince me since no persuasive proof is provided.

**Questions:**

See Weaknesses.

---

> ### Author Response · Authors · 2023-11-15
>
> Thank you for your feedback.
>
> >W1. The background is not very clear, especially for the weak signal matrix $\mathbf{W}$. For example, where does the matrix $\mathbf{W}$ come from?
>
> The matrix $\mathbf{W}$ is composed from the $m$ weak signals. Specifically, the rows of $\mathbf{W}$ are the weak signals. The $m$ weak signals are provided as an input to the problem. Based on the feedback, we have now modified the introduction so that it is clear where $\mathbf{W}$ comes from.
>
> >W2. Some preliminaries are missing. Why the expected error rate $\epsilon_i$ is equal to $\mathbb{E}\mathbf{[w_i-y]}$? Does each row of $\mathbf{W}$ kind of represent the label vector $\mathbf{y}$?
>
> We appreciate the comment. This was a typo and has been corrected in the revised version. Each row of $\mathbf{W}$ represents a weak signal that gives probabilities for data points being in a certain class. So yes, each row of $\mathbf{W}$ are weak signals, which are incomplete and noisy representations of the label vector $\mathbf{y}$.
>
> >W3. As stated in this paper, the computation of $\mathcal{H_1}$ takes time $O((nk)^{\left \lfloor{\frac{m}{2}}\right \rfloor})$ which is prohibitive in practice. The author divided $\mathbf{W}$ into several parts and took their average to reduce the value of $m$. Although the author claimed that this does not make a negative impact on the performance, it cannot convince me since no persuasive proof is provided.
>
> We agree that the performance of the method depends on the aggregation method. The computation of $\mathcal{H}_1$ is feasible when $m \leq 7$. We ran our experiments to have $m = 5$, and the performance is similar when $m$ is between $2$ to $7$. In the current work, we have not undertaken a theoretical analysis on the performance guarantees regarding the reduction of $m$. This is within the scope of future work for *RACH-Space*.

---

> > ### Comment · Reviewer_LMCF · 2023-11-20
> >
> > Thank you for your clarification. I will retain my score.

---

### Official Review · Reviewer_TWM9 · 2023-11-03

**Soundness:** 3 good
**Presentation:** 2 fair
**Contribution:** 3 good
**Rating:** 5
**Confidence:** 5

**Summary:**

The paper introduce RACH-Space for weak supervision. The method ultiizes a geometrical interpretation of weak labels and formulates the problem as a least squared problem. The paper overall has some interesting ideas and the experiments show good empirical results. However, there are some critical problems (e.g. writing, misclaims, experiment setup) that need to be addressed.

**Strengths:**

1. The paper proposes a novel method for label aggregation in weak supervision. As far as I know, the formulation seems to be new.
2. The paper provides theoretical motivations and justifications for the proposed algorithm.
3. Good experiment results are shown against baselines.

**Weaknesses:**

1. Writing can be improved. It’s better to give intuitions before presenting the equations.
2. There are some unjustified claims.
3. Experiment setup should be improved to avoid the possibility of cherrypicking.

**Questions:**

1. It’s very difficult to follow to paragraphs around equation 3.4 and 3.5. In the current form, some choices seem to be arbitrary. This is one example out from many: why do you define A=2W? Why not A=3W? I believe there are reasons and possibly principled reasons behind this, but I don’t get it from the current writing. It is mentioned in abstract that there is a geometrical interpretation. Maybe it’s helpful to introduce the geometrical interpretation first to give more intuition about the method before diving into math?
2. It seems the method heavily depends on “ average expected error rate”. How can one know average expected error rate without assuming the distribution of the error rate? How is average expected error rate defined in the paper?
3. The WRENCH dataset has 14 classification datasets, why four of the datasets were dropped in experiments? It’s the best to include all 14 datasets in order to avoid any possibility of cherry picking.
4. It is mentioned “experiments are done in the original setup on WRENCH”, but if that’s the case, how come the results are different from the results reported in the WRENCH paper? In the WRENCH paper, MeTaL had better results than MV. Is the difference caused by dropping the 4 datasets?
5. The paper claims it make minimum assumptions as MV, i.e. only assuming weak labels are better than random. However, the proposed method actually implicitly makes more assumptions. For example, it selects the average expected error rate so that b/n is in a safe region. This is already a strong assumption on that one gets to select “average expected error rate”. As the average expected error rate is expected to a single point value. Another assumption roots in the least squared problem. Why not formulate as least absolution deviation problem? What implicit assumptions are you making to formulate the problem to be a least squared problem instead of least absolution deviation?

---

> ### Author Response · Authors · 2023-11-15
>
> We appreciate the detailed feedback from the reviewer.
>
> >Q1. It’s very difficult to follow to paragraphs around equation 3.4 and 3.5. In the current form, some choices seem to be arbitrary. This is one example out from many: why do you define $A=2W$? Why not $A=3W$? I believe there are reasons and possibly principled reasons behind this, but I don’t get it from the current writing. It is mentioned in abstract that there is a geometrical interpretation. Maybe it’s helpful to introduce the geometrical interpretation first to give more intuition about the method before diving into math?
>
> In the revised version, we clarified the writing around the mentioned equations. The matrix $A$ is defined as $A=2W$. This is obtained by rearranging equation (1.2) which leads to the linear system $(2W) y=b$. Hence, the choice of $A$ being $2W$ is not arbitrary, and directly follows from the linear system. We have now added a figure and additional contexts to communicate clearly the geometrical interpretation of *RACH-Space*.
>
> >Q2. It seems the method heavily depends on “ average expected error rate”. How can one know average expected error rate without assuming the distribution of the error rate? How is average expected error rate defined in the paper?
>
> The average expected error rate is the average of the empirical expected error rates of the weak signals. Formally, $\epsilon = \frac{1}{m}\sum_{i=1}^{m} \epsilon_i$. In the revised version, we clarify that our formulation does not depend on knowing the average expected error rate. There was some confusion about the assumptions of *RACH-Space* in regards to the expected empirical error rates. In the revised version, we clarify that we do not assume prior knowledge nor a set bound for the average empirical error rates of weak signals. Instead, we update our parameter, now called pseudo-error rate, starting from $\frac{2}{k}-\frac{2}{k^2}$, which is the error rate of the weak signal that simply does random classification of the data points. *RACH-Space* does not require the average empirical error rate to take a certain value, and works independently of the average empirical error rate, which is unknown.
>
> >Q3. The WRENCH dataset has 14 classification datasets, why four of the datasets were dropped in experiments? It’s the best to include all 14 datasets in order to avoid any possibility of cherry picking.
>
> We included all 14 datasets in our updated version, and *RACH-Space* is still the best performing model compared to all existing label models. See the updated Table 3.
>
> >Q4. It is mentioned “experiments are done in the original setup on WRENCH”, but if that’s the case, how come the results are different from the results reported in the WRENCH paper? In the WRENCH paper, MeTaL had better results than MV. Is the difference caused by dropping the 4 datasets?
>
> We note that the experiments are done in the original setup on WRENCH based on the official WRENCH code from the GitHub repository. For some of the label models, our experiments yielded different results than the results on the WRENCH paper. The results we present are obtained from running the experiments on our machine. MeTaL and MV’s overall performance was not changed when the experiment was done on 10 datasets as opposed to 14 datasets.
>
> >Q5. The paper claims it make minimum assumptions as MV, i.e. only assuming weak labels are better than random. However, the proposed method actually implicitly makes more assumptions. For example, it selects the average expected error rate so that b/n is in a safe region. This is already a strong assumption on that one gets to select “average expected error rate”. As the average expected error rate is expected to a single point value. Another assumption roots in the least squared problem. Why not formulate as least absolution deviation problem? What implicit assumptions are you making to formulate the problem to be a least squared problem instead of least absolution deviation?
>
> In the revised version, we clarify that our approach is to modify $\widetilde{b}$ without requiring explicit knowledge of the average expected error rate. To make this clear, we introduce a new parameter, the pseudo-error rate. The vector $\widetilde{b}$ is initially defined using the pseudo-error rate and is subsequently updated so that it lies in the safe region. Our updates to $\widetilde{b}$ are completely algorithmic and do not rely on any concrete values of expected empirical error rates of weak signals. We made the choice of the least squares problem, since we can optimize it using standard gradient descent. We have not explored the least absolute deviation and would be happy to compare the performance of the two algorithms in terms of classification accuracy.

---

> > ### Comment · Reviewer_TWM9 · 2023-11-21
> >
> > I thank the authors for answering my questions. The revised version has improved, but I believe it is still not ready for publication considering comments from other reviewers. I decide to keep my score.

---

### Official Review · Reviewer_LHEL · 2023-11-03

**Soundness:** 2 fair
**Presentation:** 2 fair
**Contribution:** 2 fair
**Rating:** 3
**Confidence:** 2

**Summary:**

The authors introduce RACH-Space, a new classification tool within ensemble learning, ideal for situations with limited supervision (weakly supervised learning). It is easy to implement and doesn't make many assumptions about the data, using geometric interpretations of weak signal spaces. The contributions of the paper are:

1) RACH-Space, an efficient label model that provides synthetic labels for the raw dataset
2) The corresponding weakly-supervised learning algorithm
3) A theoretical analysis of the algorithm

**Strengths:**

The paper is overall well-written. Notations are correctly introduced and the relevant literature is duly cited.
The problem tackled by RACH-space is of great importance.

**Weaknesses:**

1) The contributions could be explicitly stated. It is unclear what is proved in the theoretical section.

2) Although I am not an expert in the field, I doubt the RACH-Space is particularly ground-breaking. More serious evidence on real world datasets should be added. For instance, you should test the RACH-Space algorithm on a histopathology dataset where it is common to face weakly-supervised learning problems.

**Questions:**

1) You state that "Majority Voting and RACH-Space assume that, on average, the weak labels are better than random". Could you discuss the validity of this assumption on real world datasets? Is it realistic to assume this?

---

> ### Author Response · Authors · 2023-11-15
>
> We appreciate the reviewer's feedback.
>
> >W1. The contributions could be explicitly stated. It is unclear what is proved in the theoretical section.
>
> In the revised version, the introduction now discusses the 5 contributions of the paper including the theoretical contributions.
>
> >W2. Although I am not an expert in the field, I doubt the RACH-Space is particularly ground-breaking. More serious evidence on real world datasets should be added. For instance, you should test the RACH-Space algorithm on a histopathology dataset where it is common to face weakly-supervised learning problems.
>
> All the datasets in our numerical experiments are on real world datasets. In particular, these 14 datasets are part of the WRENCH framework used as a public benchmark for weakly supervised learning on real world data.
>
> >Q1.You state that "Majority Voting and RACH-Space assume that, on average, the weak labels are better than random". Could you discuss the validity of this assumption on real world datasets? Is it realistic to assume this?
>
> We have removed the following remark on the updated version of our paper as it was no longer necessary to include, but we think that it would help in understanding the validity of this assumption.
>
> **Remark:** Naturally, a zero vector $(0, ... ,0) \in [0,1]^{nk}$ has an error rate of $\frac{1}{k}$. Thus given any set of weak signals $\mathbf{W}$, one can approximate the average expected error rate to be arbitrarily close to $\frac{1}{k}$ by concatenating lots of zero vectors as weak signals under $\mathbf{W}$. Note that $\frac{1}{k} \leq \frac{2}{k}-\frac{2}{k^2}$ where equality holds when $k=2$. Therefore, one can always modify the weak signals so that the average expected error rate is lower than $\frac{2}{k}-\frac{2}{k^2}$.
>
> Based on the remark, it can be shown that even by just adding zero vectors as weak signals, the average empirical error rate of weak signals can always be shown to be better than random. As we can always add zero vectors on real world weak signals, we think that this assumption is realistic.

---

> > ### Comment · Reviewer_LHEL · 2023-11-22
> >
> > I appreciate the authors' response to my queries. I think it still falls short of being ready for publication. Consequently, I will maintain my initial score.

---

### Official Review · Reviewer_YGuB · 2023-11-04

**Soundness:** 2 fair
**Presentation:** 1 poor
**Contribution:** 2 fair
**Rating:** 3
**Confidence:** 2

**Summary:**

Assume a dataset of size $n$ for a $k$-class classification problem is available, with unknown true labels but accompanied by $m$ independent "weak signals" that, on average, outperform random guessing. Then, the primary objective of this paper is to estimate the true labels for this weakly supervised dataset using only the mentioned $m$ independent "weak supervisors". Specifically, the authors assume that for each data point $i \in [n]$, each of these $m$ weak supervisors can assign a vector of size $k$ where the $j$th component conveys information about the probability of $X_i$ belonging to class $j$ (resulting in a total of $m\times n\times k$ signals). The paper lacks rigorous mathematical modeling regarding how this "information" is collected, and it seems that the authors rely on unspecified heuristics in this regard. From my multiple readings of this part of the paper, it appears that the mentioned weak signals somehow act as probabilities.

The proposed algorithm in this work, called RACH-space, aims to estimate the true labels by linearly combining the vectors provided by the supervisors. The authors introduce an unclear $\ell_2$-minimization scheme for determining the optimal linear combination, which may potentially yield infinitely many solutions. The remaining theoretical section concentrates on constraining the solution space and ensuring the feasibility of the estimator. Towards the end, the authors present experimental results on several datasets, although I did not thoroughly analyze this part.

The paper suffers from issues of clarity and seems to provide limited theoretical contributions. The lack of substantial mathematical modeling concerning the problem makes it challenging to assess the achievements. Many sections are difficult to follow, and it's possible that I may have overlooked critical aspects of the work. At this point, I recommend rejection. However, I kindly request the AC to seek the opinions of other reviewers with more confidence scores.

**Strengths:**

- The primary problem in this paper, i.e., learning from weakly supervised datasets, is an interesting line of work.

**Weaknesses:**

- The paper's writing quality can be significantly improved. I have identified numerous grammatical errors and vague phrases that should be addressed to enhance the paper's overall readability. Additionally, the literature review section and the section explaining the motivations behind this work lack informativeness for similar reasons.

- The authors have not provided any information or context regarding the "weak signals" in $\boldsymbol{W}$. Consequently, it is not clear why one should seek a linear combination of the rows of $\boldsymbol{W}$ to approximate the true label vector $\boldsymbol{y}$. Additionally, the motivation and the mathematical procedure leading to the optimization problem in (3.5) remain unclear.

- The theoretical contribution of this work is quite limited and may not meet the standards of ICLR. However, the authors may have achieved success from an experimental perspective, although I did not thoroughly evaluate this aspect. Therefore, if other reviewers believe that the work represents a valuable experimental contribution, I am open to reconsidering my evaluation.

**Questions:**

Please see "Weaknesses" section.

---

> ### Author Response · Authors · 2023-11-15
>
> Thank you for your feedback.
>
> >W1. The paper's writing quality can be significantly improved. I have identified numerous grammatical errors and vague phrases that should be addressed to enhance the paper's overall readability. Additionally, the literature review section and the section explaining the motivations behind this work lack informativeness for similar reasons.
>
> In the revised version of the paper, we have corrected grammatical errors and clarified phrases to improve readability. Additionally, we have rewritten the introduction to more clearly present the paper's motivation and contributions.
>
> >W2. The authors have not provided any information or context regarding the "weak signals" in $\mathbf{W}$. Consequently, it is not clear why one should seek a linear combination of the rows of $\mathbf{W}$ to approximate the true label vector $\mathbf{y}$. Additionally, the motivation and the mathematical procedure leading to the optimization problem in (3.5) remain unclear.
>
> The weak signals provide partial information on the probability that data points belong to a given class. In the revised version, we have modified the introduction and explicitly discussed how $\mathbf{W}$ is related to the true label vector $\mathbf{y}$ (see 1.1, 1.2) and motivation for the problem in (3.5).

---

> > ### Comment · Reviewer_YGuB · 2023-11-19
> > **Reponse to the authors**
> >
> > I would like to thank the authors for responding to my questions. I currently have no additional inquiries. Following a thorough review of the authors' response as well as other reviews, my stance remains that this paper requires more work before it can be considered ready for publication. Consequently, I keep my vote.

---

### Meta-Review · Area_Chair_MEB6 · 2023-12-09

**Metareview:**

This work proposes RACH-space for weakly supervised learning. The premise is that for a set of $n$ data points belonging to $k$ classes, instead of providing their labels as a set of $n$ indicator vectors in the normal supervised learning setting, one is given $m$ probability of the groundtruth labels as "weak signals".

A method is proposed to tackle this problem, but the authors provided no justification of why this would work.

**Justification For Why Not Higher Score:**

A method is proposed to tackle this problem, but the authors provided no justification of why this would work.

**Justification For Why Not Lower Score:**

N/A.

---

### Decision · Program_Chairs · 2024-01-16

Reject